

# Effects of a single bout of mobile action video game play on attentional networks

Biye Wang[1,2], Jiahui Jiang[1] and Wei Guo[1,2]

[1] College of Physical Education, Yangzhou University, Yangzhou, Jiangsu, China
[2] Institute of Sports, Exercise and Brain, Yangzhou University, Yangzhou, Jiangsu, China

## ABSTRACT

**Background**. Video game play has been linked to a range of cognitive advantages, and investigations in this domain have predominantly utilized cross-sectional designs or long-term training paradigms. Nevertheless, the specific effects of engaging in a single bout of video game play remain poorly understood. Consequently, the objective of this study is to examine the influence of a single session of mobile action video game (MAVG) play on attentional networks among college students.

**Methods**. Seventy-two nonvideo game players were assigned randomly into an MAVG and a control game group. Participants in the MAVG group engaged in a 60-minute session of an action video game played on mobile phones, while the control group played a mobile card game for the same duration. All participants completed the attentional network test (ANT), which assesses alerting, orienting, and executive control network efficiencies, before and after the intervention.

**Results**. The MAVG group had significantly improved alerting network efficiency following the intervention, compared to before ($p < 0.05$); the control game group did not. Neither executive control network efficiency nor orienting network efficiency were found to be improved by the intervention.

**Conclusion**. The present data demonstrated that a single bout of MAVG play can improve alerting network efficiency selectively in young-adult college students. MAVGs may be useful for promoting attentional function with the advantages of being accessible virtually any time and anywhere.

## INTRODUCTION

Video games have become an integral part of many people's daily lives. Some 2.6 billion people, about a third of the world's population, play video games (*Narula, 2019*). Video game play can have benefits for a variety of cognitive functions, including visual attention (*Green & Bavelier, 2006b*; *Sanchez, 2012*; *Dobrowolski et al., 2015*), task switching (*Cain, Landau & Shimamura, 2012*; *Shawn Green et al., 2012*), processing speed (*Dye, Green & Bavelier, 2009a*; *Shin et al., 2017*), memory (*Ferguson, Cruz & Rueda, 2008*; *Colzato et al., 2013*; *McDermott, Bavelier & Green, 2014*), and cognitive control (*Colzato et al., 2010*; *Glass, Maddox & Love, 2013*; *Föcker et al., 2018*). Thus, video games represent a potential educational tool to facilitate learning (*Sindre, Natvig & Jahre, 2009*; *Chang et al., 2012*; *Chang et al., 2016*), as well as a potential therapeutic medium for patients diagnosed

Corresponding author
Wei Guo, guowei@yzu.edu.cn

with attention deficit/hyperactivity disorder (ADHD) (*Crepaldi et al., 2020*), dyslexia (*Franceschini et al., 2013*), multiple sclerosis (*Bove et al., 2019*), and those experiencing clinically relevant declines in their health-related quality of life (*Wolinsky et al., 2006*).

Mobile devices, such as mobile phones, tablets, and other small mobile devices, enable people to engage in gaming in short bouts in their spare minutes in daily life. For example, college students often play mobile video games during class breaks and workers often play during their commutes. In China, there were some 516 million mobile video game (MVG) users in 2020 and more than 99 percent of video game players use mobile devices (*Statista, 2021*). Although several studies have addressed the effects of action video games on cognitive function (*Oei & Patterson, 2013*; *Oei & Patterson, 2015*; *Huang, Young & Fiocco, 2017*), studies focused on the effects of gaming on cognition have adopted mostly cross-sectional designs or long-term training paradigms. There is limited information in the literature concerning the effects of a single bout of a video game intervention on cognition, despite the prevalent engagement of a substantial populace in brief, sporadic bouts of mobile video gaming.

It is imperative to acknowledge that long-term training may not be a viable option for all individuals. Notably, not everyone may be inclined to cultivate a gaming habit. Furthermore, it is worth noting that cross-sectional studies can elucidate correlations but are inherently limited in establishing causal relationships. Exiting studies have yielded controversial conclusions regarding the short-term training effect on cognition (*Buelow, Okdie & Cooper, 2015*; *Liu et al., 2017*; *Rice et al., 2021*). Therefore, it is imperative that we gain a deeper understanding of short-term training.

Attention, a basic function that facilitates interactions with one's environment by enabling one to focus selectively on relevant information over other information, is a core process of cognition that regulates other cognitive systems. Video games have been shown to have positive effects on multiple dimensions of attention, such as attentional control (*Cain et al., 2014*; *Chisholm & Kingstone, 2015*; *Föcker et al., 2018*), visual selective attention (*Green & Bavelier, 2003*; *Green & Bavelier, 2006b*; *Karle, Watter & Shedden, 2010*; *Chisholm & Kingstone, 2012*), attention tracking (*Green & Bavelier, 2006a*; *Oei & Patterson, 2013*; *Dobrowolski et al., 2015*), and attentional resource management (*Green & Bavelier, 2003*; *Green & Bavelier, 2006b*; *Castel, Pratt & Drummond, 2005*; *Feng, Spence & Pratt, 2007*). Based on attentional network theory, the attention network test (ANT) is a validated tool for measuring three aspects of attention in a single test, namely the alerting network, the orienting network, and the executive control network (*Posner & Petersen, 1990*; *Petersen & Posner, 2012*). Previous studies employing the ANT have shown that video game players exhibit improvements in all three networks simultaneously and have associated video game play with improved functioning of the orienting and executive networks. However, the effect of a single bout of MVG play on attentional networks have not be determined to the best of our knowledge.

In the present study, we conducted a comparative analysis between a single session of mobile action video game (MAVG) play and mobile card game play, which aligns with the prevailing approach employed in the majority of intervention studies within this field (*Bavelier & Green, 2019*). Our study adhered to a conventional pre-post design, featuring

one experimental arm primarily focused on training participants in an action video game, and one active control arm involving training on a distinct commercial video game title devoid of action mechanics. The selection of these specific game genres was based on the recognition that action video games encompass gameplay elements that place a substantial demand on the attentional system, a characteristic less commonly observed in other game genres (*Bavelier & Green, 2019*). We chose to examine the effects of playing a real-time strategy (RTS) action video game, rather than a first-person shooter (FPS) game, as players of the RTS game have been shown to exhibit greater cognitive enhancement than players of the FPS video game (*Dobrowolski et al., 2015*). FPS games typically offer an egocentric perspective, which limits the number of stimuli visible to players on the screen at any given time. Conversely, RTS video games offer an allocentric viewpoint, enabling players to view a greater number of units within their field of view. As a result, RTS games necessitate players to attend to more units simultaneously, which places greater demands on their attentional processes (*Dobrowolski et al., 2015*). The allocentric viewpoint in RTS games necessitates players to maintain a high state of alertness since players are often required to monitor their enemies, screen states, or game tasks. Therefore, we adopted a popular action real-time strategy called "Honor of Kings" to investigate effects of a single MAVG bout on attentional networks. The game requires players to remain aware of a changing environment and to have to attend to hundreds of moving stimuli from an allocentric perspective. Our hypothesis is that playing a single 60-minute bout of the mobile action video game "Honor of Kings" will enhance attentional network functions when compared to the control game group, as measured by the Attention Network Test (ANT).

## MATERIALS & METHODS

### Participants

Undergraduate students were recruited by posting recruitment notices around the Yangzhou University campus. Each participant was given the Edinburgh Handedness Inventory and given video game experience and health questionnaires. There were four inclusion criteria: (1) played video games no more than 2 h per week during the past 6 months (*Glass, Maddox & Love, 2013*); (2) right-handedness (*Fan et al., 2005*; *Konrad et al., 2005*; *Neuhaus et al., 2010*; *Han et al., 2014*); (3) normal or corrected-to-normal vision; (4) no reported history of brain injury or of any psychiatric or neurological disorders. A sample size calculation was conducted by using G*Power (version 3.1), which indicated that a minimum of 54 participants would be required to achieve a power of 0.8 and detect a medium effect size of 0.25 at a significance level of 0.05. In total, A sample of 72 participants who met the inclusion criteria were then randomized into an MAVG group [N = 36, 19 females; mean age = 22.03, standard deviation (SD) = 1.73] and a control game group ($N = 36$, 17 females; mean age = 22.28, SD = 1.80).

The experimental procedure adhered to the Declaration of Helsinki and was approved by the Ethical Committee of Yangzhou University (YXYLL-2023-083). Upon arrival at the laboratory, each participant signed a written consent form.

## Attention assessments

The ANT, a validated instrument developed by *Fan et al. (2002)*, was used to assess three attention networks. The version of the ANT program employed in the present study corresponds to the one previously described by *Wang & Guo (2020)*. Stimuli were displayed on an LCD monitor that was placed at a viewing distance of 100 cm. At the beginning of each ANT trial, a fixation cross appeared in the center of the computer screen. After random time intervals in the range of 400–1,600 ms, the screen changed according to cue conditions (no cue, center cue, double cue, or spatial cue). The cue screen remained for 100 ms and cues provided information about the target stimulus in the form of one or two asterisks. In the no cue condition, the fixation cross remained alone on the screen. In the center cue condition, an asterisk replaced the fixation cross (center-middle screen). In the double cue condition, two asterisks appeared, one above and one below the fixation cross. In the spatial cue condition, a single asterisk appeared above or below the fixation cross. When the cue disappeared, the fixation item was shown alone for 40 ms.

A target stimulus consisting of five horizontally arranged Flanker arrows or lines was then presented. The visual angle for the entire set of Flanker arrows or lines is 6.8°, while that of the center arrow is 1.3°. In the spatial cue condition, the location of the target stimulus (above or below the fixation cross site) was consistent with the cue. Target consistency with the center arrow determined the cue condition. In the consistent and inconsistent conditions, the other four arrows pointed or did not point in the same direction as the central arrow, respectively. In the neutral condition, there were lines, instead of arrows, with no directional information. The target stimulus screen was closed upon the participant responding or after 1,700 ms if there was no response. The participants were asked to press the key "1" on a numeric keyboard if the arrow pointed left, and press the key "3" if it pointed right. They were instructed to press the key as quickly and accurately as possible. The ANT comprised four blocks, each consisting of 48 trials. In each block, the trials presented distinctive combinations of cue conditions, flanker conditions, arrow orientations, and target display positions. A visual overview of the task is shown in Fig. 1.

## Intervention

Participants in the MAVG group were given a brief tutorial on how to play the game, practiced operating the Honor of Kings game (Tencent Games, Shenzhen, China) for 5 min, and then played the game for 60 min on mobile phones. The duration of the single intervention in this study was determined by the prolonged battles in the game "Honor of Kings," and referencing previous studies (*Salminen & Ravaja, 2008*; *Wang et al., 2009*).

Participants in the control group engaged in a 60-minute session of Happy Poker on mobile phones, a widely recognized card game developed by Tencent Games. Although this game shares several characteristics with the experimental game, such as enjoyment and engagement, it does not involve the same cognitive constructs as the experimental game. The inclusion of an active control group was crucial to address concerns associated with non-specific placebo-based effects, as emphasized by *Bavelier & Green (2019)*. Additionally, it enabled us to ensure that all participants spent a similar amount of time using screens and experienced comparable levels of visual fatigue to a certain extent.

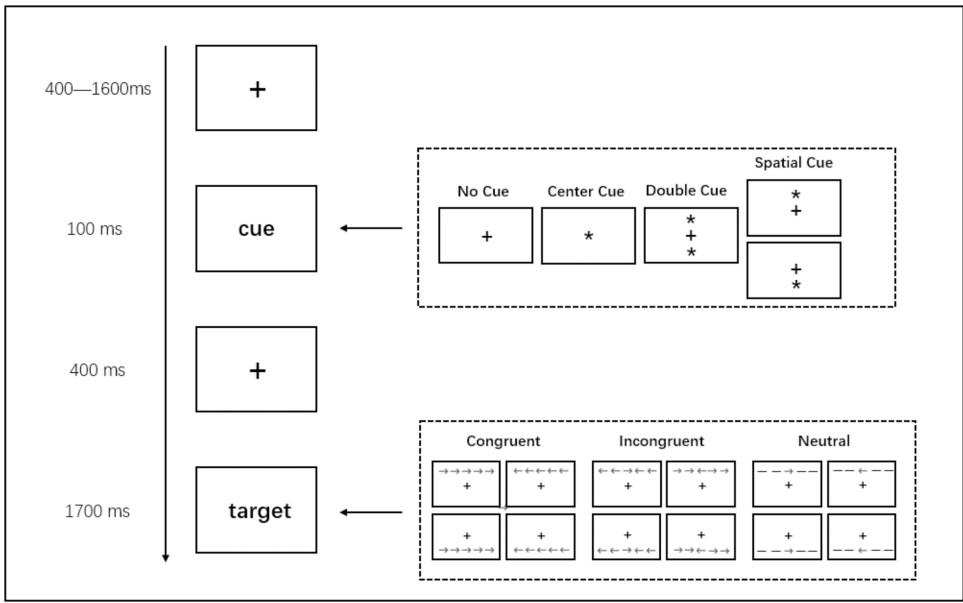

**Figure 1  Stimuli and experimental paradigm of the Attention Network Test (ANT).**

## Procedure

One week before the experiment, participants who met the recruitment criteria were contacted individually by telephone to confirm their video game play experience; they were informed of the time and place of the experiment if they met the requirements. Participants were asked to abstain from playing MAVGs and from consuming caffeine-containing substances or alcohol for 24 h prior to the experiment.

Upon their arrival at laboratory, the participants signed a consent form were given game instructions. After confirming that they understood the instructions, the eligible participants began the ANT separately in a quiet and dimly lit after understanding the instructions. A practice session with 24 trials preceded the experiment. Participants were required to obtain 80% accuracy in the practice session before commencing with the experiment. They were allowed to repeat the practice block again if they did not achieve 80% accuracy in the first attempt.

Upon successful completion of the practice session, participants were allowed to start the experimental task, (four blocks, 48 trials in each block). Between the blocks, they could rest and then initiate the next block by pressing any key. After finishing, they were assigned randomly to either the MAVG or control game group for the intervention session after a short break. Both groups' participants completed post-intervention ANT 10 min after the intervention in order to minimize any transient effect of heightened arousal levels (*Kasuya-Ueba, Zhao & Toichi, 2020*; *Yu et al., 2020*; *Ishihara et al., 2021*).

## Design and statistical analysis

The present study adopted a 2 (GROUP: MAVG, Control game) × 2 (TIME: Pre- and Post-intervention) mixed-factor design. GROUP was a between-subjects variable while

TIME was a within-subject variables. The dependent variables included reaction times (RTs), accuracy in the attentional network test, and the efficiency of each attentional networks.

The first statistical analysis in examining these networks consisted of conducting a 2 (GROUP) × 2 (TIME) × 3 (FLANKER TYPE) × 4 (CUE TYPE) mixed factors analysis of variance (ANOVA) on RTs and accuracy data, respectively. FLANKER TYPE (Incongruent, Congruent, Neutral) and CUE TYPE (Central Cue, Spatial Cue, Double Cue, No Cue) were within-subject variables.

Then, efficiency of each attentional networks (Alerting, Orienting, and Executive Control) was subjected to a 2 (TIME) × 2 (GROUP) repeated-measures ANOVA. Alerting, orienting, and executive control efficiencies were computed as No-cue RT minus Double-cue RT, Central-cue RT minus Spatial-cue RT, and Incongruent-flanker RT minus Congruent-flanker RT, respectively. Due to the three distinct large-scale neural networks underlying different aspects of attention (*Posner & Petersen, 1990*; *Petersen & Posner, 2012*), the three networks were analyzed separately as previous studies (*Sun et al., 2012*; *Gold et al., 2013*; *Fu et al., 2018*; *Xia et al., 2022*). The criterion for significance was set at $p < 0.05$.

Raw data used in this article can be found at Supplemental Information 1.

## RESULTS

### Demographic characteristics

The MAVG and control game groups were confirmed to be similar with respect to age, t (70) = 0.60, $p = 0.55$, amount of time spent playing video games per week in the past 6 months, t (70) = 1.53, $p = 0.13$. Furthermore, there were no differences between the groups in terms of basic response times, which were calculated based on response times across all conditions, t (70) = 1.28, $p = 0.21$.

### Mean RTs

For the RT analysis, all incorrect trials or trials that were three SDs from the individual mean were excluded. An ANOVA revealed significant main effects of CUE TYPE ($F_{(3, 210)} = 319.54$, $p < 0.01$, $\eta_p^2 = 0.82$) and FLANKER TYPE ($F_{(2, 140)} = 1,345.52$ $p < 0.01$, $\eta_p^2 = 0.95$). Post hoc comparisons using the least significant difference (LSD) method indicated that participants exhibited significantly faster response times in the spatial cue condition (Mean = 442.13, SE = 3.44) compared to the no cue (Mean = 486.05, SE = 3.72, $p < 0.01$), center cue (Mean = 468.44, SE = 3.71, $p < 0.01$), and double cue (Mean = 462.35, SE = 3.60, $p < 0.01$) conditions. Regarding FLANKER TYPE, participants responded significantly faster in the congruent condition (Mean = 441.68, SE = 3.54) compared to the incongruent (Mean = 508.93, SE = 3.86, $p < 0.01$) and neutral (Mean = 453.62, SE = 3.42, $p = 0.03$) conditions.

Furthermore, we observed a significant interaction between FLANKER TYPE and CUE TYPE ($F_{(6, 420)} = 9.17$, $p < 0.01$, $\eta_p^2 = 0.12$). We conducted a FLANKER by CUE ANOVA, contrasting congruent, incongruent, and neutral conditions. Interaction analysis revealed that participants response significant faster in the congruent trails than in the

**Table 1 Group mean RTs (ms) and SDs by cue and flanker type before and after the intervention.**

| Flanker type | Cue Type | MAVG Group | | Control Game Group | |
|---|---|---|---|---|---|
| | | *Pre* | *Post* | *Pre* | *Post* |
| Congruent | *No Cue* | 473.18 ± 50.11 | 464.83 ± 52.58 | 461.61 ± 42.31 | 463.17 ± 31.67 |
| | *Center Cue* | 446.23 ± 48.67 | 438.32 ± 50.45 | 436.84 ± 39.01 | 445.24 ± 36.51 |
| | *Double Cue* | 443.60 ± 48.12 | 427.71 ± 41.82 | 438.98 ± 34.46 | 440.10 ± 36.68 |
| | *Spatial Cue* | 429.10 ± 50.20 | 415.59 ± 41.42 | 418.00 ± 39.69 | 424.45 ± 28.14 |
| Incongruent | *No Cue* | 529.37 ± 46.46 | 520.37 ± 57.78 | 529.92 ± 45.76 | 528.67 ± 38.64 |
| | *Center Cue* | 529.86 ± 43.85 | 507.31 ± 53.93 | 521.30 ± 50.78 | 514.97 ± 40.81 |
| | *Double Cue* | 517.67 ± 45.66 | 495.59 ± 53.40 | 513.87 ± 48.54 | 508.75 ± 37.22 |
| | *Spatial Cue* | 489.52 ± 44.72 | 466.74 ± 48.33 | 484.78 ± 45.34 | 484.14 ± 41.78 |
| Neutral | *No Cue* | 468.64 ± 45.17 | 464.94 ± 49.97 | 464.07 ± 36.22 | 463.80 ± 35.80 |
| | *Center Cue* | 448.72 ± 38.45 | 439.85 ± 48.84 | 446.43 ± 34.11 | 446.24 ± 35.87 |
| | *Double Cue* | 446.00 ± 43.15 | 427.13 ± 40.61 | 445.07 ± 44.91 | 443.74 ± 35.56 |
| | *Spatial Cue* | 428.92 ± 48.12 | 415.97 ± 47.47 | 420.74 ± 38.14 | 427.61 ± 40.41 |

incongruent trails in the no cue ($F (1, 143) = 631.93$, $p < 0.01$, $\eta_p^2 = 0.82$), central cue ($F (1, 143) = 942.75$, $p < 0.01$, $\eta_p^2 = 0.87$), double cue ($F (1, 143) = 860.36$, $p < 0.01$, $\eta_p^2 = 0.86$), and spatial cue ($F (1, 143) = 633.42$, $p < 0.01$, $\eta_p^2 = 0.82$) conditions. Significant differences between incongruent and neutral trials were also found in the no cue ($F (1, 143) = 697.52$, $p < 0.01$, $\eta_p^2 = 0.83$), central cue ($F (1, 143) = 903.45$, $p < 0.01$, $\eta_p^2 = 0.86$), double cue ($F (1, 143) = 842.41$, $p < 0.01$, $\eta_p^2 = 0.86$), and spatial cue ($F (1, 143) = 538.24$, $p < 0.01$, $\eta_p^2 = 0.79$) conditions. Specifically, the response speed of participants significantly faster in the neutral trials compared to the incongruent in all cue conditions. Congruent trials differed significantly from neutral trials only in the center cue condition ($F (1, 143) = 4.04$, $p = 0.046$, $\eta_p^2 = 0.03$), the response speed in the congruent trials was significant faster than the neutral trails in the center cue condition. No significant differences were observed in the no-cue ($F (1, 143) = 0.03$, $p = 0.87$, $\eta_p^2 < 0.01$), double cue ($F (1, 143) = 2.16$, $p = 0.14$, $\eta_p^2 = 0.02$), and spatial cue ($F (1, 143) = 0.68$, $p = 0.41$, $\eta_p^2 < 0.01$) conditions.

Additionally, we detected significant TIME × FLANKER TYPE interactions ($F (2, 140) = 3.95$, $p = 0.02$, $\eta_p^2 = 0.05$). A TIME by FLANKER ANOVA comparing pre- and post-intervention tests showed that RTs were significantly shorter after the intervention under the incongruent condition ($F (1, 287) = 9.26$, $p < 0.01$, $\eta_p^2 = 0.03$), with no significant differences observed in the congruent ($F (1, 287) = 1.11$, $p = 0.29$, $\eta_p^2 = 0.04$) and neutral ($F (1, 287) = 2.32$, $p = 0.13$, $\eta_p^2 < 0.01$) conditions. No other main effects or interactions reached statistical significance. Descriptive data for these conditions can be found in Table 1.

## Accuracy

Mean accuracy values for the groups are reported with SDs in Table 2. An ANOVA revealed significant main effects on accuracy of CUE TYPE ($F (3, 210) = 9.97$, $p < 0.01$, $\eta_p^2 = 0.13$) and FLANKER TYPE ($F (2, 140) = 149.08$, $p < 0.01$, $\eta_p^2 = 0.68$), but not of GROUP ($F$

**Table 2  Group mean accuracy (%) and SDs by cue type and flanker type before and after the intervention.**

| Flanker type | Cue Type | MAVG Group | | Control Game Group | |
|---|---|---|---|---|---|
| | | *Pre* | *Post* | *Pre* | *Post* |
| Congruent | *No Cue* | 97.92 ± 4.48 | 97.92 ± 3.66 | 96.88 ± 5.07 | 97.92 ± 3.66 |
| | *Center Cue* | 99.64 ± 1.45 | 97.74 ± 6.00 | 99.13 ± 2.19 | 98.96 ± 2.80 |
| | *Double Cue* | 99.13 ± 2.19 | 98.26 ± 4.62 | 98.78 ± 2.51 | 99.13 ± 2.19 |
| | *Spatial Cue* | 98.96 ± 2.36 | 97.05 ± 4.60 | 98.09 ± 4.68 | 99.13 ± 3.39 |
| Incongruent | *No Cue* | 89.93 ± 10.05 | 89.58 ± 10.35 | 89.76 ± 8.34 | 91.32 ± 9.12 |
| | *Center Cue* | 91.49 ± 9.70 | 88.54 ± 11.23 | 89.41 ± 9.43 | 91.84 ± 7.14 |
| | *Double Cue* | 90.45 ± 9.27 | 89.41 ± 11.46 | 90.10 ± 7.82 | 90.45 ± 10.19 |
| | *Spatial Cue* | 94.79 ± 5.88 | 92.53 ± 7.29 | 93.23 ± 7.96 | 94.27 ± 6.91 |
| Neutral | *No Cue* | 97.40 ± 4.32 | 97.22 ± 4.59 | 97.05 ± 3.80 | 97.40 ± 3.77 |
| | *Center Cue* | 98.09 ± 3.28 | 97.22 ± 4.59 | 98.61 ± 3.03 | 99.48 ± 1.75 |
| | *Double Cue* | 98.26 ± 4.12 | 97.74 ± 4.00 | 99.13 ± 2.19 | 98.09 ± 3.60 |
| | *Spatial Cue* | 98.78 ± 2.51 | 98.09 ± 3.60 | 98.96 ± 3.50 | 98.96 ± 2.80 |

$(1, 70) = 0.69$, $p = 0.41$, $\eta_p^2 = 0.01$) or TIME ($F (1, 70) = 0.24$, $p = 0.63$, $\eta_p^2 < 0.01$). Subsequent post hoc comparisons employing the least significant difference (LSD) method for CUE TYPE revealed that response accuracy in the no cue condition was significantly lower than in the center cue ($p = 0.04$) and spatial cue ($p < 0.01$) conditions. Notably, response accuracy in the spatial cue condition was significantly higher than in the center cue ($p < 0.01$) and double cue conditions ($p < 0.01$). No significant differences were observed between the no cue and double cue conditions or between the center cue and double cue conditions. Regarding FLANKER TYPE, participants exhibited significantly higher accuracy in the congruent and neutral conditions compared to the incongruent condition ($p < 0.01$), while no significant difference was found between the congruent and neutral conditions ($p = 0.21$).

Furthermore, the ANOVA for accuracy revealed a significant CUE TYPE × FLANKER TYPE interaction ($F (6, 420) = 6.55$, $p < 0.01$, $\eta_p^2 = 0.86$). We conducted a FLANKER by CUE ANOVA, contrasting congruent, incongruent, and neutral conditions. Interaction analysis demonstrated that the accuracy of congruent trials significantly higher than incongruent trails in the no cue ($F (1, 143) = 79.65$, $p < 0.01$, $\eta_p^2 = 0.36$), central cue ($F (1, 143) = 112.20$, $p < 0.01$, $\eta_p^2 = 0.44$), double cue ($F (1, 143) = 119.25$, $p < 0.01$, $\eta_p^2 = 0.46$), and spatial cue ($F (1, 143) = 52.64$, $p < 0.01$, $\eta_p^2 = 0.27$) conditions. Additionally, he accuracy was significantly higher in the neutral trials compared to the incongruent in the no cue ($F (1, 143) = 84.20$, $p < 0.01$, $\eta_p^2 = 0.37$), central cue (F(1, 143) = 108.47, $p < 0.01$, $\eta_p^2 = 0.43$), double cue ($F (1, 143) = 94.61$, $p < 0.01$, $\eta_p^2 = 0.40$), and spatial cue ($F (1, 143) = 81.49$, $p < 0.01$, $\eta_p^2 = 0.36$) conditions. However, no significant difference between congruent and neutral trials was found in no cue ($F (1, 143) = 0.77$, $p = 0.38$, $\eta_p^2 < 0.01$), central cue ($F (1, 143) = 1.91$, $p = 0.17$, $\eta_p^2 = 0.43$), double cue ($F (1, 143) = 1.68$, $p = 0.20$, $\eta_p^2 = 0.40$) and spatial cue ($F (1, 143) = 0.95$, $p = 0.33$, $\eta_p^2 = 0.36$) conditions. No other interactions reached significant level.

## Attentional network efficiencies

An ANOVA for alerting network efficiency showed a significant main effect of TIME ($F$ (1, 70) = 5.39, $p < 0.05$, $\eta_p^2 = 0.07$) and GROUP ($F$ (1, 70) = 5.97, $p = 0.02$, $\eta_p^2 = 0.08$). *Post-hoc* comparisons indicated that the alerting network efficiency of the MAVG group (Mean = 27.27, SE = 2.07) was significantly higher than that of the control group (Mean = 20.12, SE = 2.07). Furthermore, the post-intervention alerting network efficiency scores (Mean = 27.13, SE = 2.01) were significantly higher than the scores obtained in the pre-intervention assessment (Mean = 20.26, SE = 2.15). A significant TIME × GROUP interaction ($F$ (1, 70) = 3.94, $p < 0.05$, $\eta_p^2 = 0.06$) was observed for alerting network efficiency. To provide a more detailed analysis of alerting network efficiency in response to this interaction, we conducted a GROUP by TIME ANOVA, comparing the MAVG and control groups. No significant between-group differences were found in the pre-intervention test ($F$ (1, 35) = 0.27, $p = 0.61$, $\eta_p^2 < 0.01$). However, significant between-group differences emerged in the post-intervention test ($F$ (1, 35) = 7.80, $p < 0.01$, $\eta_p^2 = 0.08$). The alerting network efficiency of the MAVG group (Mean = 33.24, SE = 2.84) was significantly higher than that of the control group (Mean = 21.01, SE = 2.84) in the post-intervention test (Fig. 2).

In contrast, an ANOVA for orienting network efficiency yielded no significant main effect of TIME ($F$ (1, 70) = 0.003, $p = 0.96$, $\eta_p^2 < 0.01$), GROUP ($F$ (1, 70) = 0.49, $p = 0.49$, $\eta_p^2 < 0.01$) and no significant TIME × GROUP interaction ($F$ (1, 70) = 1.61, $p = 0.21$, $\eta_p^2 = 0.02$).

Finally, an ANOVA for executive network efficiency showed a significant main effect of TIME ($F$ (1, 70) = 5.00, $p = 0.03$, $\eta_p^2 = 0.07$). Post-intervention executive network efficiency scores (Mean = 71.09, SE = 2.46) were significantly lower than the pre-intervention scores (Mean = 63.39, SE = 2.34, $p < 0.05$). However, no significant main effect of GROUP ($F$ (1, 70) = 2.25, $p = 0.14$, $\eta_p^2 = 0.03$) and no significant TIME × GROUP interaction ($F$ (1, 70) < 0.01, $p > 0.99$, $\eta_p^2 < 0.01$) was detected.

## DISCUSSION

The present study utilized the popular action real-time strategy game known as "Honor of Kings" as a means to examine the impact of a short-term mobile action video game (MAVG) play on attentional networks. The findings revealed that a single 60-min bout playing MAVG could improve attentional networks, particularly of the alerting network, in a sample of young-adult college students generally considered to be at the peak of their cognitive functions (*McArdle et al., 2002*; *Bialystok, Craik & Luk, 2012*; *Pérez et al., 2014*). This result is consistent with previous studies that have suggested that gaming may improve alerting network efficiency (*Colzato et al., 2013*; *Fowler & Gustafson, 2019*). The observed improvement could be consequent to the strategy game adopted in the present study, which requires players to make decisions quickly in real-time against an opponent. Participants are required to switch between a more diffused attentional state (monitoring for enemies) and a focused attentional state (engaging enemies). This dynamic interplay places considerable demands on attentional systems and heightens visual

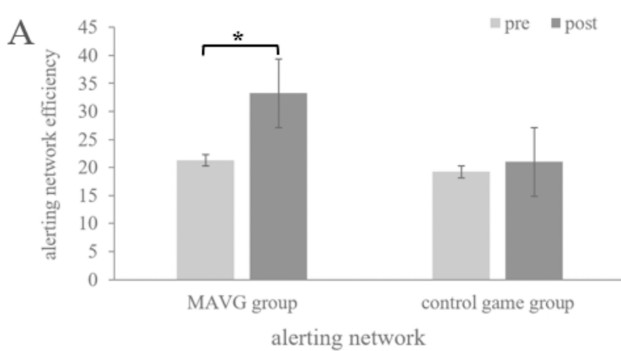

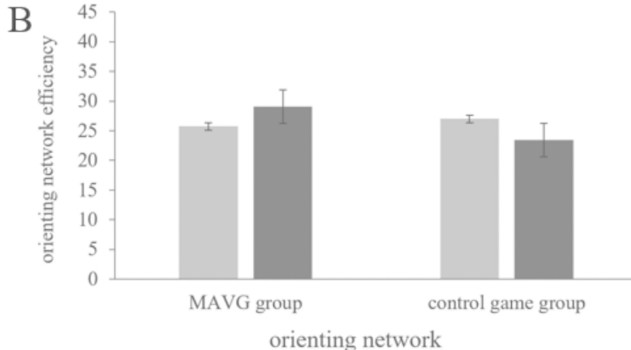

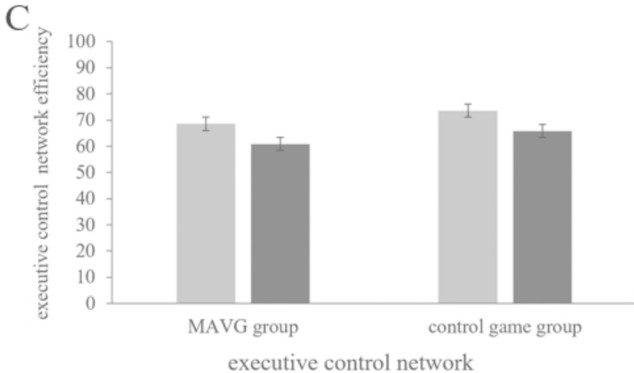

**Figure 2  Efficiency of the alerting (A), orienting (B) and executive control network (C) in mobile game and control groups.** ($^*p < 0.05$). Alerting, orienting, and executive control efficiencies were computed as follows: alerting—no-cue RT minus double-cue RT, orienting—central-cue RT minus spatial-cue RT, and executive control—incongruent-flanker RT minus congruent-flanker RT.

sensitivity, contributing to the observed improvements (*Li et al., 2009*). Previously, overall enhancement in mobile real-time strategy gamers has been shown to be related more to faster information accumulation than changes in motor response mappings (*Green, Pouget & Bavelier, 2010*; *Mack, Wiesmann & Ilg, 2016*). During each bout of MAVG, players need to maintain a high state of arousal and alertness, which could underlie augmentation of the efficiency of the alerting network. One of the underlying mechanisms may be attributed to the effects of video games on pupil diameter and velocity, resulting in an increase in norepinephrine release (*Fowler & Gustafson, 2019*). Previous studies on both humans and

monkeys have revealed that norepinephrine release has a positive impact on alertness (*Aston-Jones & Cohen, 2005*). Concurrently, individuals who engage in action video games exhibit distinct neural patterns, including reduced activation in the fronto-parietal network and enhanced recruitment of the temporal parietal junction, in contrast to those who do not partake in such gaming activities. These neural patterns mirror their superior ability to flexibly allocate attention and effectively filter out distractions (*Bavelier et al., 2012*; *Föcker et al., 2018*). Consequently, it is reasonable to hypothesize that the observed enhancement in alertness within the MAVG group may be attributed to these underlying factors.

Interestingly, our MAVG did not exhibit improvement in the efficiencies of the orienting and executive networks, which was inconsistent with previous reports (*Boot et al., 2008*; *Dye, Green & Bavelier, 2009b*; *Li et al., 2019*). There are several potential explanations for these inconsistent results. Firstly, the effects of a playing on a mobile gaming platform may differ from the effects of playing on a console/computer gaming platform, which were used in previous studies (*Glass, Maddox & Love, 2013*; *Dobrowolski et al., 2015*). Plausibly, disparities may emerge in terms of cognitive demands and the engagement of cognitive domains across these distinct gaming platforms, potentially leading to varying impacts on cognitive functions. Worth noting is a recent study, which, despite failing to demonstrate statistically significant differences in cognitive function attributable to platform selection, did illuminate a pronounced association between choice of platform and distinct preferences for action video game genres, as well as the underlying motivations that drive individuals to partake in gaming activities (*Huang, Young & Fiocco, 2017*). Secondly, we tested the effects of a single gaming session, with the aim of examining short-term effects. Prior research has predominantly employed long-term intervention design to uncover effects stemming from prolonged gaming experiences (*i.e.,* ≥10 h of gameplay), necessitating rigorous and extended training sessions (*Green & Bavelier, 2006a*; *Green & Bavelier, 2006b*; *Bailey & West, 2013*; *Kühn et al., 2014*). In previous studies, there may have been some cognitive advantage of long-term video games by way of skill transfer from video games and cognitive tasks (*Oei & Patterson, 2014*; *Dobrowolski et al., 2015*). The effects of a single MAVG bout on attention has rarely been discussed. Thirdly, it is possible that video game genre may influence the results. We adopted a real-time strategy game (a specific game genre), whereas game genre was not specified explicitly in previous studies (*Boot et al., 2008*; *Dye, Green & Bavelier, 2009b*; *Shawn Green et al., 2012*; *Li et al., 2019*). Playing action video games of different genres may have differential enhancing effects on cognitive functions (*Dobrowolski et al., 2015*).

The present results suggest the MAVGs may, in addition to being a leisure activity, be useful for promoting attentional function with the advantages of being accessible virtually any time and anywhere. Alerting network deficiencies have been demonstrated in patients with ADHD and in HIV-infected patients (*Johnson et al., 2008*; *Wang et al., 2017*). The present results suggest that MAVGs may have potential to improve alerting network function, particularly in these populations. Indeed, the video game Endeavor Rx was approved in the USA as a prescription for the treatment of children with ADHD (*Kollins et al., 2020*). Moreover, professions that demand heightened levels of vigilance may explore MAVGs as a means of combating fatigue. For instance, air traffic control,

where attentiveness is crucial, could potentially leverage MAVGs for this purpose (*Fowler & Gustafson, 2019*).

There were several limitations in the present study. First, we observed a higher level of accuracy in the ANT compared to prior studies (*Dye, Green & Bavelier, 2009b*; *Wang, Guo & Zhou, 2016*). While this heightened accuracy implies a strong level of participant engagement, which augments the credibility and reliability of our findings, it is plausible that a ceiling effect may have been encountered. Further exploration may establish its efficacy in assessing attentional networks. It is worth noting that the recently introduced video game-like assessment tool, Attention Trip, has been posited as a potentially more engaging alternative to the original ANT paradigm (*Klein et al., 2017*; *Klein, Vallis & Chisholm, 2019*). Secondly, we conducted a phenomenological analysis that cannot provide insight into the underlying mechanisms of selectively enhancing an attentional network. The mechanism by which a single session of gaming can affect attention needs to be further explored. Thirdly, since the current study is a single session intervention study, it is not possible to know whether the improvements in alerting networks could be maintained over time.

# CONCLUSIONS

The present study has contributed evidence supporting the potential benefits of a single bout MAVG on attentional network function. Our observations revealed that a single 60-minute MAVG session resulted in a discernible enhancement in the efficiency of the alerting network, while other networks remained unaffected. These findings suggest that MAVGs may serve not only as a leisure activity but also as a practical means to enhance attentional function. Importantly, the accessibility of MAVGs at virtually any time and location further underscores their utility in promoting attentional capabilities.

## Funding

This work was supported by the Higher School Natural Science Foundation of Jiangsu Province (No. 23KJB320023). The funders had no role in study design, data collection and analysis, decision to publish, or preparation of the manuscript.

## Grant Disclosures

The following grant information was disclosed by the authors:
Higher School Natural Science Foundation of Jiangsu Province: 23KJB320023.

## Competing Interests

The authors declare there are no competing interests.

## Author Contributions

- Biye Wang conceived and designed the experiments, analyzed the data, prepared figures and/or tables, authored or reviewed drafts of the article, and approved the final draft.

- Jiahui Jiang performed the experiments, analyzed the data, prepared figures and/or tables, authored or reviewed drafts of the article, and approved the final draft.
- Wei Guo conceived and designed the experiments, authored or reviewed drafts of the article, and approved the final draft.

### Human Ethics

The following information was supplied relating to ethical approvals (i.e., approving body and any reference numbers):

The experimental procedure was approved by the Ethical Committee of Yangzhou University

### Data Availability

The raw measurements are available in the Supplementary File.

### Supplemental Information

Supplemental information for this article can be found online at http://dx.doi.org/10.7717/peerj.16409#supplemental-information.

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
