# Peer review of "Effects of a single bout of mobile action video game play on attentional networks"

_PeerJ, doi:10.7717/peerj.16409_

## Round 0.1 · original submission · Major Revisions

Thank you for your submission. We have now received two reviews, which are presented below. In line with both reviewers' suggestions, I have chosen to request major revisions to this manuscript.

As you will see, there is some overlap in the reviewers' comments, but also numerous independent points. Please could you be careful to address all comments by both reviewers before resubmitting your manuscript?
Thanks again for submitting to PeerJ, and all the best with your revisions.

Reviewer 1 ·

Basic reporting

The manuscript has been clearly written, professional English has been used throughout the study.
Literature: The discussion could point to further literature: Possible underlying principles of differences in performance have been discussed by referring to eye tracking tasks. However, underlying differences could also be explained by neural attentional network changes similarly as observed in action video game players. I recommend integrating the possible underlying neural mechanisms as one further principle in the discussion. These are outlined in: Bavelier et al., 2012; Foecker et al., 2018.

Bavelier, D., Achtman, R. L., Mani, M., & Föcker, J. (2012). Neural bases of selective attention in action video game players. Vision research, 61, 132-143.

Föcker, J., Cole, D., Beer, A. L., & Bavelier, D. (2018). Neural bases of enhanced attentional control: Lessons from action video game players. Brain and behavior, 8(7), e01019.

Figures could be improved by exactly including labels of the dependent variable (efficiency is not clear).
A table of the exact games being played during the past 6 months and prior to the past 6 months (if possible should be included)

I suggest to include a Table (in the Appendix) including the types of video games which have been played during the 6 months for each participant.
This is because of the following situation:
Criteria: (1) played video games no more than 2 hours per week during the past 6 months (Glass, Maddox & Love, 117 2013).
These criteria are not very strict *during the past 6 months, as there might be people who played much more games prior to these 6 months. Could the authors check and include a table of the video games played and possible durations?

Raw data have been shared, but they are not well explained, it is not clear what group 1 or 2 is (at least I did not see any explanations). I would also include a link in the manuscript (OSF).



Stimulus Material: Can the visual angle of the material be described?
How were participants instructed to perform the task (as fast and as correct as possible)?
Open Science: Could the authors include an open science statement in their manuscript?

Experimental design

The research question is well defined: 60 Minutes MOBA training, the research is "new" in the context of a 60 minutes MOBA training, however the dependent variable (Attentional network task) is a standard paradigm, and in the context of the training, it would be informative to also record data during the training, as well as investigate gender differences.


Stimulus Material: Can the visual angle of the material be described?
How were participants instructed to perform the task (as fast and as correct as possible)?

Validity of the findings

Results: there is not much evidence for group differences (only one significant interaction has been reported).


Subordinate analysis (comparisons) of main effects and interactions - post hoc t-tests should be reported

Conclusions could also point to future research

Additional comments

My complete review can be found below:

In this training experiment, the authors tested the impact of a Multiplayer Online Battle Arena (MOBA) game training on reaction times, accuracy and efficiencies recorded from the attentional network task. The attentional network task consists of three different types of conditions:
• alerting (cue),
• orienting (spatial cue) and
• executive control (flanker task).
The experimental conditions consisted of different types of cues:
When the cue was absent, the fixation cross was presented throughout, whereas in the double cue condition, two asterisks have been presented one above and one below fixation. Spatial cues were presented, by introducing a single asterisk either above or below fixation. Cue presentation was followed by a target in the spatial cue condition which appeared at the location indicated by the spatial cue.
Additionally, consistent (compatible) and inconsistent (incompatible) conditions have been presented, in which the central arrow pointed to the same direction as the other arrows or in which the central arrow pointed into the opposite direction of the surrounding arrows. In the no cue condition, only a central arrow was presented without any surrounding flanker. Participants were asked to respond to the target.
72 participants (36 in the training group (19 females) and 36 participants in the control group (17 females)) were trained for one hour on either a mobile action video game (MOBA) or a mobile card game (control game). Pre-and post-training performance was analysed by running an ANOVA including the factors Time (pre, post training), Group (MOBA versus card game), Cue Type (no cue, double cue, spatial cue, central cue) and Flanker Type (Congruent, Incongruent, Neutral). The results showed a faster response times in the double and central cue condition in the experimental group compared to the control group. Accuracy did not show any group differences. Alerting network efficiency differed significantly between the pre- and posttest intervention tests for the training group. The authors conclude that “ the alerting network” is efficiently improved in young adults.

While this study contributes to the understanding of the impact of MOBA training on the attentional network tasks, there are several methodological questions and suggestions for improving the discussion:
Methods:
I suggest to include a Table (in the Appendix) including all types of video games which have been played during the 6 months for each participant.
This is because of the following situation:
Criteria: “There were four inclusion criteria: (1) played video games no more than 2 hours per week during the past 6 months (Glass, Maddox & Love, 117 2013).
These criteria are not very strict *during the past 6 months, as there might be people who played much more games prior to these 6 months. Could the authors check and include a table of the video games played?

Stimulus Material: Can the visual angle of the material be described?
How were participants instructed to perform the task (as fast and as correct as possible)?
Open Science: Could the authors include an open science statement in their manuscript?


Results:
Were there any gender differences? As the authors recruited an almost equal amount of males and females this question could be further addressed.
Did the authors record any data during game play?
Could the authors report the results of the subordinate analysis (e.g. the Group by Cue Type interaction)? (e.g. main effects and interactions?)


Figures: could the authors re-lable the y achsis (exact ependent variable)?


Discussion: Possible underlying principles have been discussed by referring to eye tracking tasks, as well as the noradrenergic system. However, underlying differences could also be explained by neural attentional network changes similarly as observed in action video game players. I recommend integrating the possible underlying neural mechanisms as one further principle in the discussion. These are outlined in: Bavelier et al., 2012; Foecker et al., 2018.
Bavelier, D., Achtman, R. L., Mani, M., & Föcker, J. (2012). Neural bases of selective attention in action video game players. Vision research, 61, 132-143.
Föcker, J., Cole, D., Beer, A. L., & Bavelier, D. (2018). Neural bases of enhanced attentional control: Lessons from action video game players. Brain and behavior, 8(7), e01019.

Minor: Lines 239-240:
Wording of the sentence – please remoce “was found in alerting network”
Include citation complete citation: (Fan et al.

Reviewer 2 ·

Basic reporting

I have reviewed the manuscript titled "The Effect of a Single Bout of Video Game Training on Attentional Network". The study aims to investigate the impact of a single session of video game training on the attentional network. The authors have conducted a comprehensive review of relevant previous studies and identified a significant research gap in the existing literature. Overall, the manuscript is well-structured and effectively communicates the research findings.

However, I have identified several points that the authors may consider addressing to enhance the clarity of the writing:
Line 51 and 52, consider removing the “as well as” and merging the last condition, e.g., “multiple sclerosis (Bove et al, 2019), and those who have experienced an extensive decline in their health-related quality of life (Wolinsky et al., 2006).”
From Line 62, the authors started to talk about the research gap in previous studies. Consider using a separate paragraph to introduce the gap and elaborate a bit on why we need to address this gap. This could highlight the reason and rationale of this study.
Line 75-77, the sentence, “The attentional test paradigms used in these studies including a multiple object tracking task, an attentional blink test, an enumeration task, a useful field of view task, a flanker compatibility task, and a letter detection task.”, does not contribute to the storyline. Consider deleting this. Otherwise, if attentional paradigms matter, please specify this.
Line 88. Consider replacing “pre-test, training, post-test” with “pre-post”.
Line 104. Consider merging this paragraph into the last one.
Line 129. Rectify “Fan et al. (Fan et al.)”
Line 153. The authors introduced the single bout training in the procedure. Consider using a new subheading (e.g. intervention) to introduce the intervention separately.
Line 179. Wrong formatting, “as emphasized by Bavelier and Green (2019) (Bavelier & Green, 2019)”.
Line 252, before discussing the result, may consider adding a short opening paragraph to restate the aim and summarize the main findings.
Line 257, “The observed improvement could be consequent to the strategy game adopted in the present study requiring players to make decisions quickly in real-time against an opponent and to switch between a more diffused attentional state (monitoring for enemies) and a focused attentional state (engaging enemies), which would be expected to place loads on attentional systems and increase visual sensitivity (Li et al., 2009).” The sentence is hard to read and ambiguous. Consider short sentences instead.
Line 277-279, previous studies suggest that mobile gaming is better. Then why there is a limited effect in this study while traditional gaming platform reported borad effects on attention?
Line 279, the second explanation is superficial, only stating the difference in playing session may result in this difference. Please consider what could be the reasons for this. E.g., learning requires intensive training sessions?
Line 292, consider framing this point as a limitation
Line 322, consider adding some implications to the conclusion.

Experimental design

This study used a rigor 2 by 2 experimental design. ParticulaIt this study set up an appropriate control group, which can rule out many alternative explanations for results caused by gaming. Besides, the attentional tasks used in this study are standard. The sample size was well established.

However, the introduction of outcomes is not clear. Please elaborate on the dependent variables in each task and how the dependent variables were calculated if necessary.

Validity of the findings

The authors used a proper and standard data analysis. However, the authors did not report how the simple effects were conducted following the interaction.

Additional comments

There has been extensive research on the effects of multi-session video gaming on attention; however, this study contributes to the field by addressing the gap - how single-session video gaming can affect attention. With the use of a regor experimental design, I believe this study can raise some awareness and stimulate further research in this area.

---

## Round 0.2 · Minor Revisions

As you can see, R2 is now satisfied with the manuscript and the changes you have made. However, R1 still has some minor revisions that they would like to see incorporated into the final manuscript. I agree with R1 in terms of enhancing the clarity in your reporting of the stats, the inclusion of effect sizes, etc. I hope you can address these comments to everyone's satisfaction, and I look forward to receiving your revised submission.

Reviewer 1 ·

Basic reporting

The report is clearly written, professional English has been used throughout. The suggested/recommended literature has been included in the discussion. Raw data have been shared.

Experimental design

The experimental design has been clearly described and presented.

Validity of the findings

The research includes impact and novelty. Underlying data have been provided

Additional comments

The authors have answered and included my comments. However,I still have some comments on the result section outlined below:

Results: Demographic characteristics:
The authors write:
"The MAVG and control game groups were confirmed to be similar with respect to age, t (70) = 0.60, p = 0.55, amount of time spent playing video games per week in the past 6 months, t (70) = 1.53, p = 0.13, and basic response time, t (70) = 1.28, p = 0.21."
- It is not clear how response times were calculated here. Are these response times across all conditions? Could the authors specify which data they used to calculate the basic response time?

Line 246ff: The authors write: "Post hoc comparisons using the least significant difference (LSD) method indicated that participants exhibited significantly faster response times in the spatial cue condition compared to the no cue (p < 0.01), center cue (p < 0.01), and double cue (p < 0.01) conditions."
Here, the authors could indicate the means and standard deviation or errors for the spatial, centre and double cue. Alternatively, the authors could refer to the table in which these descriptive results have been reported.

Lines 254-263: The authors write: "Interaction analysis revealed significant differences between congruent and incongruent trials in the no cue (F (1, 143) = 631.93, p < 0.01), central cue (F (1, 143) = 942.75, p < 0.01), double cue (F (1, 143) = 860.36, p < 0.01), and spatial cue (F (1, 143) = 633.42, p < 0.01) conditions . Significant differences between incongruent and neutral trials were also found in the no cue (F (1, 143) = 697.52, p < 0.01), central cue (F (1, 143) = 903.45, p < 0.01), double cue (F (1, 143) = 842.41, p < 0.01), and spatial cue (F (1, 143) = 538.24, p < 0.01) conditions. Congruent trials differed significantly from neutral trials only in the center cue condition (F (1, 143) = 4.04, p = 0.046), with no significant differences observed in the co-cue (F (1, 143) = 0.03, p = 0.87), double cue (F (1, 143) = 2.16, p = 0.14), and spatial cue (F (1, 143) = 0.68, p = 0.41) conditions."

It would be more informative to report the direction of these effects. For instance if participants were faster in condition x than y. I also recommend to include means and standard errors for these different conditions or refer to the table in which this information has been presented.

Lines 296 ff: The authors write: "We conducted a FLANKER by CUE ANOVA, contrasting congruent, incongruent, and neutral conditions. Interaction analysis demonstrated significant differences between congruent and incongruent trials in the no cue (F (1, 143) = 79.65, p < 0.01), central cue (F (1, 143) = 112.20, p < 0.01), double cue (F (1, 143) = 119.25, p < 0.01), and spatial cue (F (1, 143) = 52.64, p < 0.01) conditions. Additionally, significant differences between incongruent and neutral trials were observed in the no cue (F (1, 143) = 84.20, p < 0.01), central cue (F(1, 143) = 108.47, p < 0.01), double cue (F (1, 143) = 94.61, p < 0.01), and spatial cue (F (1, 143) = 81.49, p < 0.01) conditions. However, no significant difference between congruent and neutral trials was found in no cue (F(1, 143) = 0.77, p = 0.38), central cue (F(1, 143) = 1.91, p = 0.17), double cue (F(1, 143) =1.68, p = 0.20) and spatial cue (F(1, 143) =0.95, p = 0.33) conditions."

I recommend to use Partial eta squared consistently throughout the result section.

Lines 322-324: The authors write: "No significant between-group differences were found in the pre-intervention test (F(1, 35) = 0.27, p = 0.61). However, significant between-group differences emerged in the post-intervention test (F(1, 35) = 7.80, p < 0.01)".

I recommend to indicate the direction here, I would indicate here explicitly which group showed better performance and either report mean and SEs in the text or refer to the table.

For Figures: I recommend to specify the description of the y achsis further and give more explanations here,

Reviewer 2 ·

Basic reporting

The authors have addressed the concerns I had previously raised.

Experimental design

The authors have addressed the concerns I had previously raised.

Validity of the findings

The authors have addressed the concerns I had previously raised.

Additional comments

The authors have addressed the concerns I had previously raised.

---

## Round 0.3 · accepted · Accept

Thank you for addressing the reviewers' final comments, and I am happy to accept your revised manuscript for publication. Congratulations! And thank you for submitting to this journal. All the best, Robin.